# Early Life Stress and High FKBP5 Interact to Increase Anxiety-Like Symptoms through Altered AKT Signaling in the Dorsal Hippocampus

**DOI:** 10.3390/ijms20112738

**Published:** 2019-06-04

**Authors:** Marangelie Criado-Marrero, Niat T. Gebru, Lauren A. Gould, Taylor M. Smith, Sojeong Kim, Roy J. Blackburn, Chad A. Dickey, Laura J. Blair

**Affiliations:** Department of Molecular Medicine, USF Health Byrd Institute, University of South Florida, Tampa, FL 33613, USA; marangelie@health.usf.edu (M.C.-M.); niat@health.usf.edu (N.T.G.); lgould1@health.usf.edu (L.A.G.); taylormarina@mail.usf.edu (T.M.S.); sojeongkim@mail.usf.edu (S.K.); rjblackburn@brandeis.edu (R.J.B.); cdickey@health.usf.edu (C.A.D.)

**Keywords:** FKBP5, early life stress, anxiety, hippocampus, AKT

## Abstract

Clinical studies show a significant association of childhood adversities and FK506-binding protein 5 (FKBP5) polymorphisms on increasing the susceptibility for neuropsychiatric disorders. However, the mechanisms by which early life stress (ELS) influences FKBP5 actions have not been fully elucidated. We hypothesized that interactions between ELS and high FKBP5 induce phenotypic changes that correspond to underlying molecular changes in the brain. To test this, we exposed newborn mice overexpressing human FKBP5 in the forebrain, rTgFKBP5, to ELS using a maternal separation. Two months after ELS, we observed that ELS increased anxiety levels, specifically in mice overexpressing FKBP5, an effect that was more pronounced in females. Biochemically, Protein kinase B (AKT) phosphorylation was reduced in the dorsal hippocampus in rTgFKBP5 mice, which demonstrates that significant molecular changes occur as a result of ELS when FKBP5 levels are altered. Taken together, our results have a significant impact on our understanding mechanisms underlying the gene x environment interaction showing that anxiety and AKT signaling in the hippocampus were affected by the combination of ELS and FKBP5. An increased knowledge of the molecular mechanisms underlying these interactions may help determine if FKBP5 could be an effective target for the treatment of anxiety and other mood-related illnesses.

## 1. Introduction

Mental health disorders are very common, affecting around 1.1 billion people worldwide and 44.7 million adults in the United States [1]. Although there is no population completely resilient to these disorders, there is a lower prevalence in men (15%) compared to women (21.7%) and a lower prevalence in older adults (14.5%) compared to young adults (22.1%), according to the 2017 National Survey on Drug Use and Health [1]. In adolescents, anxiety disorders are the most prevalent mental health conditions [2,3]. Clinical studies have highlighted several factors like age, genetics, and environment [4,5], which can increase susceptibility to neuropsychiatric diseases, including anxiety, depression, and posttraumatic stress disorder (PTSD) [6,7]. These factors are not entirely separate but can actually influence each other. It is now known that environmental factors can modify the genes implicated in mental health disorders, including genes regulating learning, memory, and emotion. We also know that some of these environmental factors, such as early life adversities, can induce long-lasting epigenetic modifications (e.g., DNA methylation) that affect gene and protein expression throughout life and, in some cases, can even be passed down through generations [5]. Some of the most commonly affected genes in neurotransmission and stress response include the serotonin transporter, brain-derived neurotrophic factor, nerve growth factor, and glucocorticoid receptor [8]. These genes encode for proteins involved in crucial developmental, synaptic, and stress response processes.

An impaired stress response, which is mediated by the hypothalamic-pituitary-adrenal (HPA) axis, is a common feature in individuals who are more susceptible to developing mental disorders [9,10]. After stress, the HPA axis mediates the secretion of glucocorticoids (cortisol in humans and corticosterone in rodents) that bind to mineralocorticoid and glucocorticoid (GR) receptors throughout the brain. Activation of these receptors serves as a negative feedback loop controlling or terminating the cellular stress response [11]. Imbalance in this feedback loop at any stage can result in short- and long-term detrimental effects in the brain, inducing neuronal death, slowed neurogenesis, weakened synaptic connections, and increased inflammation, as well as impaired learning and memory processes, which was recently reviewed [12]. One key regulator for reducing the affinity of GR for glucocorticoids is the 51 kDa FK506-binding protein, also known as *FKBP5*/FKBP51. Increased FKBP5 protein expression and common polymorphisms in the *FKBP5* gene are significantly associated with GR resistance [13]. In addition, the interaction of early life stressors (e.g., childhood abuse) with several common *FKBP5* allelic variations has been shown to increase susceptibility of many mental health disorders [14,15], indicating that dysregulation of *FKBP5* may contribute to a maladaptive stress response in these patients. One specific example, the *FKBP5* rs1360780 T-allele variant significantly increase susceptibility to depressive disorders, while the CC allelic variant has been shown to offer protection [13]. In this study, patients with major depression carrying the T-allele carriers presented reduced FKBP5 mRNA induction and higher GR resistance when compared to the FKBP5 rs1360780 CC protective genotype. In a separate study, carriers of the rs1360780 risk variant were more susceptible to anxiety and other mental health disorders when exposed to maltreatment as a child [15,16]. This variant increases *FKBP5* expression following stress [16]. However, the causal relationship between altered *FKBP5* and early life stress (ELS) has not been tested.

Here, we examined the interaction between ELS and FKBP5 in vivo using the recently developed rTgFKBP5 mouse model, which overexpresses human *FKBP5* throughout the forebrain. This model does not show any anxiety-like or depression-like phenotypes basally [17], but we hypothesized that the interaction of high *FKBP5* and ELS may yield neuropsychiatric-like symptoms through the altered stress-feedback pathways in the brain. We combined behavioral, molecular, and histochemical approaches to address this question, examining changes in the hippocampus and the amygdala, which are two main brain regions responsible for emotional and cognitive functions and known to be susceptible to stressors. We observed that anxiety levels were increased by the combination of FKBP5 and ELS. We also examined Protein kinase B (AKT) phosphorylation, since it has been previously demonstrated to be inhibited by FKBP5 [18] and is known to regulate autophagy, cell survival, and hippocampal synaptic plasticity, affecting learning and memory processes [19,20]. We found that rTgFKBp5 mice presented reduced AKT Ser374 phosphorylation in the dorsal hippocampus. Together, this result suggests that altered FKBP5 levels can combine with ELS to uniquely alter molecular pathways and increase anxiety-like behavior in vivo.

## 2. Results

### 2.1. Early Life Stress Selectively Increases Anxiety-Like Behavior in rTgFKBP5 Mice

To examine the outcomes of the interaction between an adverse environment during early ages with high FKBP5, we evaluated phenotypic and biochemical changes in mice following maternal separation as the ELS (Figure 1A). First, we evaluated anxiety levels using an elevated plus maze (EPM) in the control (WT and tTA) and FKBP51-overexpressing (rTgFKBP5) littermates. We used both the wild-type (WT) and CamKIIα-tTA (tTA) controls to ensure any effect of high FKBP51 was independent of either background. CamKIIα-tTA mice express the tetracycline-controlled transactivator protein regulated by the forebrain-specific calcium-calmodulin-dependent kinase II promoter. Although the genotype presented a significant effect on the ELS group (Table 1), the number of factors being studied reduced the significant power in the Bonferroni post-hoc test (*p* > 0.05). Additionally, sex significantly contributed to the variability (*p* = 0.05) in the ELS group. However, due the small representation of animals per sex in each group, we ad-hoc used an independent sample t-test comparing the groups. In this analysis, the rTgFKBP5-ELS mice displayed increased anxiety-like behavior as measured by a significant decrease in time in the open arms compared to tTA-ELS (t(18) = 3.489, *p* = 0.002) and WT-ELS (t(18) = 2.261, *p* = 0.036) mice (Figure 1B), which was not affected by the number of entries to open arms (Table 1) or locomotor differences among the groups (Figure 1C). rTgFKBP5-ELS mice also had significantly lower percent number of open arm entries compared to tTA-ELS mice by two-way Analysis of variance (ANOVA) (Appendix A). No significant difference was found within the non-stressed control groups (Table 1). However, sex significantly interacted with genotype (*p* < 0.05) affecting anxiety-like behavior. Overall, the rTgFKBP5-ELS female mice showed the highest level of anxiety-like behavior when compared to tTA-ELS (t_(8)_ = 3.436, *p* = 0.008) and WT-ELS (t_(8)_ = 2.807, *p* = 0.023) mice, while changes in anxiety in males varied by genotype (Figure 1D,E).

### 2.2. Sex and Genotype Modestly Affect the Prepulse Inhibition Response

Since sensorimotor gating reflexes are known to be reduced in patients with various neuropsychiatric disorders linked to *FKBP5* and stress [21], we tested for any deficiencies in startle habituation (response after repeated presentations of a stimulus) and the prepulse inhibition (PPI, response after previous presentation of a weak stimulus). Startle habituation was similar within all groups (Figure 2A). However, a significant difference within the ELS group was measured by PPI, but only following the 78 dB stimulus (Figure 2B,C). Once again, we found sex differences between the groups. Specifically, the stressed rTgFKBP5 females had lower inhibition of startle (*p* < 0.05) when compared to the WT and tTA controls (Appendix A). Overall, we only found modest changes in the overall startle response and PPI caused by the interaction of ELS and *FKBP5* (*p* > 0.05) (Table 1).

### 2.3. FKBP5 Induces Spatial Reversal Learning Deficits in the Morris Water Maze (MWM) Test

We previously demonstrated that rTgFKBP5 mice have impaired spatial learning and cognitive flexibility using the MWM reversal task [17]. Here, we examined whether ELS could intensify this impairment. We found no difference between the groups in the initial training or probe test (Appendix A), except that the rTgFKBP5-ELS mice spent significantly more time in the target quadrant during the Day 5 probe trial than the tTA-ELS, WT-ELS, and non-stressed rTgFKBP5 (Bonferroni post-hoc, *p* < 0.05). While this result is interesting, all of the groups were able to successfully identify the target quadrant. To test for cognitive flexibility, we relocated the platform in the opposite quadrant. Despite a similar escape latency for the reversal learning among the groups during the reversal training (Figure 3A,C), learning deficits were apparent during the reversal probe test (Figure 3B,D). The rTgFKBP5 mice were unable to identify the new target quadrant spending similar time among the all quadrants, corroborating our previous findings [17]. However, this effect was not enhanced by ELS (Bonferroni post-hoc, *p* > 0.05).

### 2.4. Early Life Stress Affects the pAKT/AKT Ratio in the Hippocampus

Levels of FKBP51 have been previously demonstrated to alter AKT signaling in vitro. Therefore, we wanted to examine how this important signaling cascade was affected by the interaction of FKBP51 and ELS. We measured AKT activity by quantifying the expression of phosphorylated AKT at Serine 473 (pAKT^Ser473^, active form) and total AKT as well as the phosphorylation status of Glycogen synthase kinase 3 beta (GSK-3β), which is directly downstream in the AKT signaling pathway, in the whole hippocampus (HPC). We found that pAKT^Ser473^ was significantly increased in the HPC by ELS, as determined by two-way ANOVA (Figure 4A,B), but GSK-3β was not affected throughout the HPC (*p* > 0.05) (Figure 4C). Post-hoc analysis of the pAKT^Ser473^ levels showed no difference among the groups (*p* > 0.05). To look more closely at how this pathway is regulated by *FKBP5*, we examined two hippocampal subregions that have different functions and present differential stress sensitivity [22].

### 2.5. FKBP5 Differentially Regulates AKT Phosphorylation in the Dorsal Hippocampus in the Presence Or Absence of ELS

Traditionally, the hippocampus has been studied for memory functions. However, the dorsal (DH) and ventral (VH) areas of the hippocampus differ in their inputs and projections to different cortical and subcortical structures [23]. In both hippocampal subregions, we assessed whether genotype and ELS affect AKT and GSK-3β levels. Again, we evaluated changes in the active form of AKT (pAKTSer473), which may subsequently prevent GSK-3β activation by phosphorylating this kinase at serine 9. We first confirmed that, in the dorsal hippocampus, rTgFKBP5 mice presented higher levels of FKBP5 than the WT and tTA mice (genotype effect: *p* < 0.0001) (Figure 5A). Then, a two-way ANOVA revealed a significant interaction in stress and genotype in the pAKT^Ser473^/AKT ratio (*p* = 0.006) (Figure 5B). In the non-stressed group, rTgFKBP5 had a significantly lower pAKT^Ser473^/AKT ratio compared to WT and tTA (Bonferroni post-hoc, *p* < 0.05) mice. These results are in line with previous cell culture studies showing that increased expression of FKBP5 reduces AKT phosphorylation [19,24]. Interestingly, ELS significantly reverted the pAKT^Ser473^/AKT ratio in rTgFKBP5-ELS mice when compared to non-stressed-rTgFKBP5 mice (Bonferroni post-hoc, *p* < 0.05). Since GSK-3β is a downstream target of AKT, we expected lower levels of Serine 9 phosphorylated GSK-3β (pGSK-3β^Ser9^, inactive form) as a result of having less pAKT^Ser473^ in non-stressed rTgFKBP5 mice. However, the pGSK-3β^Ser9^/GSK-3β ratio was not affected by the interaction of stress and genotype in the DH (Figure 5C,D).

### 2.6. AKT and GSK-3β Expression in the VH and Amygdala (AMYG) is not Affected by the FKBP5 x ELS Interaction

We further examined if AKT and GSK3β were changed in the ventral hippocampus. We found that even though rTgFKBP5 mice showed high expression of FKBP5 (*p* < 0.0001) (Figure 5E), the pAKT^Ser473^/AKT ratio was unaffected by genotype, stress, or interaction of both (Figure 5F). Similar results were observed in the pGSK-3β^Ser9^/GSK-3β ratio, which showed no main effects by genotype, stress, or their interaction (Figure 5G and H). Comparable to the VH findings, the overexpression of FKBP5 in the AMYG (Figure 5I) did not affect the pAKT^Ser473^/AKT ratio (Figure 5J). Similarly, the pGSK-3β^Ser9^/GSK-3β ratio (Figure 5K,L) remained unchanged in the AMYG suggesting that the interaction of FKBP5 and stress was not implicated in the increased anxiety in rTgFKBP5-ELS mice.

## 3. Discussion

Preclinical and clinical studies have demonstrated that the *FKBP5* rs1360780 risk allele increases the susceptibility to develop mental health illnesses when combined with adverse environments during early ages [6,7,14,16,25,26,27]. Therefore, in this study, we investigated the relationship between ELS and high FKBP5 in the brain. We used a unique mouse model overexpressing FKBP5 (rTgFKBP5) and two control groups (WT and tTA) to address our hypothesis. The tTA control group express the tetracycline-off transactivator driven by the CAMKII promoter, which helped us distinguish rTgFKBP5 mice from any phenotypical and biochemical difference caused by this construct. We found that stress significantly exacerbated anxiety levels in rTgFKBP5-ELS mice but did not alter spatial reversal learning. Besides the behavioral and cognitive alterations, the rTgFKBP5 mice showed a reduction of phosphorylated AKT at Ser473 in the dorsal hippocampus, but this was increased in the ELS group. This demonstrates that genotype alone can induce direct molecular changes that are further regulated by ELS. Overall, this suggests that the interaction of gene and environment may cause long term neurobiological changes, which may be relevant to patients with neuropsychiatric disorders.

### 3.1. Sex as a Confounding Factor that Effects Stress Susceptibility

Interestingly, the rTgFKBP5 females presented the highest anxiety levels out of all the groups. This finding adds to the growing number of studies examining both sexes, in rodents and humans, that demonstrate how sex can contribute to differences in brain development, biological regulation, and emotional reactivity throughout the lifespan [28,29,30,31,32]. To our interest, clinical studies have reported higher anxiety prevalence and poor brain connectivity in women who were previously exposed to trauma and carriers of *FKBP5* risk variants [33,34]. In addition to elevated FKBP5 levels, other factors, such as hormones and activity of steroid receptors, may also impact anxiety levels in females. Estrogen levels during the reproductive cycle (estrous in mice and menstrual in women) can also affect stress responsiveness, anxiety, fear neurocircuits, and activation of brain structures, potentially contributing to sex differences [35,36]. Although our goal in this study was to test any interaction between stress and FKBP5 expression, for future studies it would be interesting to examine how estrogen and steroid receptors are affected by *FKBP5* and stress, which might directly or indirectly contribute the observed sex differences in anxiety. It is important to note that previous studies have demonstrated significant differences in female rats using EPM [37], which may be applicable to other rodents. It will be valuable to confirm these using other anxiety-like behavioral tests in future studies. Anxiety-like behaviors, as many other behaviors, are difficult to assess in rodents. This difficulty arises from the number of mice needed to observe significant differences, the inclusion of sex as an independent variable in previous studies to make valuable comparisons, and the limited information about the biochemical and physiological differences between sexes in rodents’ brain.

### 3.2. Early Life Stress Modestly Interacted with Genotype 

Although we were anticipating that ELS would cause robust behavioral and molecular effects in all genotypes, this stress had only a modest impact on just the rTgFKBP5 mice. It is possible that we could have achieved a stronger effect by applying a secondary stressor before behavioral testing, which is supported by the three-hit hypothesis [38]. Further studies are needed to test this alternative and examine if the combination of early- and later-life stressors induces more severe changes in the brain. Additionally, some behavioral and molecular discrepancies have been reported based on mouse strain (35), so another possibility is that our ELS protocol (maternal separation) was not sufficient to induce significant changes in startle response or enhanced cognitive deficits in our Friend Virus B (FVB) background mice. It is also possible that more phenotypic and biochemical changes would be detectible at an older age.

### 3.3. Genotype Influences Learning Processes at the Behavioral and Molecular Levels

We previously showed that rTgFKBP5 mice had impairments in spatial learning and cognitive flexibility while promoting the α-amino-3-hydroxy-5-methyl-4-isoxazolepropionic acid, AMPA, receptor recycling in hippocampal neurons [17]. However, ELS did not alter this cognitive impairment in rTgFKBP5, so we did not follow-up on synaptic changes in the present study. Instead, we decided to examine changes in the AKT/GSK-3β signaling, which is known to be affected by FKBP5 [39]. We found only minor changes in the levels of these kinases by immunohistochemistry of the whole hippocampus. Interestingly, the tTA group appears to have a non-significant increase in pAKT levels (Figure 4). This may be due to the regulation of the CAMKIIα transgene, since CAMKIIα has been previously demonstrated to regulate AKT [40]. The presence of high FKBP51 appears to reduce this, but additional mice are needed to determine if this reaches significance. Biochemically, our findings revealed that high FKBP5 significantly decreases pAKT^Ser473^/AKT ratio in vivo (Figure 5). This is in line with previously in vitro reports [19,39]. Interestingly, this reduction in the pAKT^Ser473^/AKT ratio was only observed in the dorsal, but not ventral, hippocampus in rTgFKBP5 mice. Another previous study showed that overexpression of FKBP5 in the dorsal hippocampus did not have an effect on anxiety-like behaviors. However, different from our study, their investigation only included males, and the behavioral testing was performed at an older age than in our experiments [41].

### 3.4. FKBP5 Overexpression Modulates AKT in Selective Brain Structures

In addition to spatial learning deficits in rTgFKBP5 mice, we observed differences in anxiety levels (Figure 1B). Despite the extensive literature about the role of the hippocampus in mood disorders, there are limited studies examining the role of FKBP5 and stress affecting AKT in hippocampal axes. Here, we investigated the AKT activity in the dorsal hippocampus, which is mainly involved in cognitive processes like spatial learning and in the ventral hippocampus which plays a greater role modulating emotional information such as anxiety. One key finding in this study is the reduction of pAKT^Ser473^ in the dorsal hippocampus of the rTgFKBP5 mice. Considering that overexpression of GSK-3β impairs spatial memory formation [42], we suspected that FKBP5-induced reduction of pAKT^Ser473^ in the dorsal hippocampus would lead to increased GSK-3β activity, but this was not the case. Similar results were also reported in another study [24], suggesting that AKT may use other intermediary proteins that could be effected by FKBP5 to modulate GSK-3β.

The pAKT^Ser473^ changes were only observed in the dorsal hippocampus, but not in the ventral axis nor the amygdala. This outcome was unexpected since the ventral hippocampus and amygdala, which are highly susceptible to stress, are key structures regulating memory and emotional processes [43,44]. More recently, a study showed that induction of FKBP5 in the basolateral (BLA) and central (CeA) nuclei of the amygdala caused an anxiety-like behavior in male mice supporting its role in modulating susceptibility to anxiety disorders [41]. One possibility for not observing changes in the AKT pathway may be that the tissue analyzed contained various amygdala nuclei. Due their dissociable functions of these amygdala nuclei in anxiety [45] and other mental health disorders [46], if this was the case, the difference may not be detectible. Additional studies are needed to address specific changes in these amygdalar subregions.

### 3.5. Concluding Remarks

Our findings highlight the importance of stress and genes (like FKBP5) in modulating vulnerability to mood disorders and learning impairments. This study demonstrates how sex may contribute to susceptibility and prevalence differences in these illnesses and cognitive processes. Our data revealed that changes in the AKT signaling in the dorsal hippocampus may be contributing to the reduced cognitive flexibility exhibited in the rTgFKBP5 mice during the Morris Water Maze reversal test. One limitation of our study is lacking an appropriate number of animals representing each sex per each group which prevent us from making conclusions about possible sex influence on molecular findings. We will also need this information to establish any possible correlation between behavior and molecular outcomes. Besides sex differences, it will be interesting to further study how ELS differentially affects the hippocampal axes’ synaptic plasticity in rTgFKBP5 mice. This information could contribute to our knowledge on how cognitive and emotional components may influence hippocampal plasticity and activity after stressful events. These changes may also inform us about long-lasting neuronal changes affecting our susceptibility to stress at later ages. Taken together, our results support our hypothesis and is align with many clinical studies suggesting that the interaction of high FKBP5 x stress could contribute to the development of mental health disorders.

## 4. Materials and Methods

### 4.1. Animal Subjects

Mice were obtained from our colony at the University of South Florida (USF) vivarium. They were housed up to five per cage, maintained under standard conditions with a 12 h light/dark cycle, and had free access to food and water. All animal experiments were carried out accordingly with the National Institutes of Health (NIH) Guide for the Care and Use of Laboratory Animals and approved on the 15th of November 2016 [R2543] by the USF Institutional Animal Care and Use Committee (IACUC).

Generation of the rTgFKBP5 mice has been previously described in [17]. Briefly, rTgFKBP5 single-transgenic founder mice on a FVB background were crossed with CAMKIIα-tTA (tTA) mice on the 129S6 background (internally maintained colony originally a gift from Jada Lewis, strain can be purchased from Jackson Labs, stock 003010). The offspring corresponds to the rTgFKBP5 double-transgenic line. All genotypes (WT, tTA, rTgFKBP5) were confirmed by polymerase chain reaction (PCR) amplifying DNA from ear clips using the following primers: (1) human FKBP5 gene (F, 5′GTGTACGGTGGGAGGCCTAT3′, and R, 5′GTCCCATGCCTTGATGACTT3′) and (2) housekeeping gene T-cell receptor delta chain (Tcrd) (F, 5′-CAAATGTTGCTTGTCTGGTG-3′, and R, 5′-GTCAGTCGAGTGCACAGTTT-3′). A QIAxcel Advanced system (Qiagen, Valencia, CA, USA) was used to analyze the PCR product.

### 4.2. Early Life Stress (ELS) Model

We used the maternal separation protocol, which is a widely used procedure for early life stressors in rodents [47,48]. During the ELS protocol, mice underwent maternal separation for 3 h per day for 14 days, starting on postnatal day 1 (P1) (Figure 1A). This separation consists of relocating the dams to new cages in a separate room. We used six groups to examine changes induced by stress, genotype, or interaction between both. We used both WT and tTA littermates (expressing the tetracycline-off transactivator driven by the CAMKII promoter) as controls. The non-stressed control group remained undisturbed with their dams until they were weaned (P21). Although this is considered a chronic stress model, no pups were lost during the maternal separation procedure. Over the five-month experiment, no overt physical abnormalities were observed in any of the mice.

### 4.3. Elevated Plus Maze (EPM)

The EPM, as previously described [31], consisted on four perpendicular arms (two open and two closed) with an elevation of 40 cm off the floor. Each mouse was individually placed in the center is the platform and allowed to explore for 5 min. Exploration of center, closed arms, and open arms was tracked and recorded using ANY-maze software (www.anymaze.com). The chamber was cleaned with 10% ethanol between trials.

### 4.4. Prepulse Inhibition (PPI)

PPI was performed as previously described [31]. Briefly, mice were individually placed in a clear Plexiglass restrainer located inside a SR-LAB Startle apparatus (San Diego Instruments, San Diego, CA, USA). Each session was initiated with a 5 min acclimation exposing the mice to constant background white noise. This was followed by a 120 dB startle stimulus. This 120 db startle stimulus with no prepulse was used to measure the baseline. The prepulse set consisted of 5 different acoustic stimuli at 74, 78, 82, 86, and 90 dB. Trials were separated by a randomized inter-trial interval (ITI) of 10 and 30 s. PPI was calculated as the percent inhibition of the startle response from baseline, as measured by the maximum velocity of the movement in a set window following the acoustic stimulus.

### 4.5. Morris Water Maze (MWM)

MWM was performed as previously described [17]. The first day, mice were placed in random quadrants for a total of 4 trials in a large circular pool allowing them to find a visible platform that was placed in random locations around the pool within 60 s period (Appendix A). Then, for the next four days, mice were given a total of 4 trials of 60 s each to locate a hidden platform in an open pool (learning phase) with spatial cues. Each mouse was allowed to remain on the platform for 15 s at the end of each trial. The escape latency, or time it took to find the platform, was recorded by a blind observer. When the animal failed to find the platform within the allotted time, it was manually placed on the platform and allowed to remain for 15 s. To assess spatial memory, mice were exposed to one 60 s trial (probe) twenty-four hours after completing the learning phase. Swimming time in the target quadrant versus the opposite, adjacent right, and adjacent left quadrants was used as a measure for spatial memory. For the spatial reversal learning, the acquisition and test trials were repeated as described above, but the hidden platform was placed in the opposite quadrant. The results from acquisition trials were averaged across 4 trials per day (mean ± S.E.M.). ANY-maze video tracking software was used to record and analyze the videos.

### 4.6. Tissue Collection, IHC, and Western Blot

These procedures are described in more detail in our previous publication [1]. Briefly, one week after behavioral testing, 5-month-old mice were euthanized with a Somnasol overdose and transcardially perfused with 0.9% saline. In half of the cohort [Non-stressed control (WT= 6, tTA = 5, rTgFKBP5 = 5), ELS (WT= 5, tTA = 5, rTgFKBP5 = 5)], we collected the brains where the left hemisphere was transferred to 4% paraformaldehyde for overnight fixation and the right hemisphere was used to isolate brain structures. Free-floating 25 µm sections were used for immunohistochemical analysis. Tissue sections were incubated overnight with the following primary antibodies: anti-pAKT^Ser473^ (CS-4060; 1:1000) and anti-GSK-3β (CS-9832; 1:1000). A Zeiss AxioScan.Z1 slide scanner (Zeiss, Oberkochen, Germany ) was used to image 20 x brightfield sections followed by tissue cytometry using the NearCYTE software (www.nearcyte.org).

In the remaining animals (Non-stressed control (WT= 5, tTA = 5, rTgFKBP5 = 4), ELS (WT= 5, tTA = 5, rTgFKBP5 = 5)), the whole brain was collected after 0.9% saline perfusion and 1 mm brain tissue punches were obtained from the hippocampus (dorsal & ventral) and amygdala. Tissue was homogenized in radioimmunoprecipitation assay buffer, RIPA, buffer containing phosphatase and protease inhibitors. Tissue lysates were placed in a shaker at 4 °C for 30 min and then centrifuged at 10,000× g for 15 min. Protein quantification in the supernatant was performed using a bicinchoninic acid assay, BCA, assay (Thermo Scientific, #PI-23225, Waltham, MA, USA). A total of 25 µg of sample was loaded into a Sodium dodecyl sulfate polyacrylamide gel electrophoresis (SDS–PAGE) gel and ran at 120 V until the dye front was near the bottom of the gel. Proteins were then transferred to a PVDF membrane. Ponceau staining was performed to measure total protein. Membranes were blocked using 7% milk and then incubated overnight with a blocking buffer containing the following antibodies one at a time: anti-FKBP5 (CS-8245; 1:1000), anti-pAKT^Ser473^ (CS-4060; 1:1000), anti-AKT (CS-9272; 1:1000), anti-pGSK-3β^Ser9^ (CS-9323; 1:1000), anti-GSK-3β (CS-9832; 1:1000), anti-GAPDH (Proteintech, 10494-1AP; 1:1000, Rosemont, IL, USA). Images were taken using the GE ImageQuant LAS 4000 imager (Piscataway, NJ, USA) and protein bands were analyzed using the Scion Image software (www.scion-image.software.informer.com).

### 4.7. Statistical Analysis

A total of non-stressed (WT= 11, tTA =10, rTgFKBP5 = 9) and maternal-stressed (WT= 10, tTA = 10, rTgFKBP5 = 10) mice were used per group for behavioral testing. Each group contained a balanced number of males (~5) and females (~5) (Figure 1A). Interaction between genotype and ELS was examined by a two-way ANOVA analysis using the SPSS program. A two-way repeated measures ANOVA was performed for PPI and MWM behavioral outcomes. Protein expression was analyzed with two-way ANOVA (PRISM, Graph Pad Software; https://www.graphpad.com). ANOVA was followed by Bonferroni correction to compared individual means within the experimental group or genotypes. A difference was considered significant if *p* < 0.05. Data are presented as mean ± standard error of the mean (SEM). All statistical analyses can be found in Table 1 and Table 2.

## Figures and Tables

**Figure 1 ijms-20-02738-f001:**
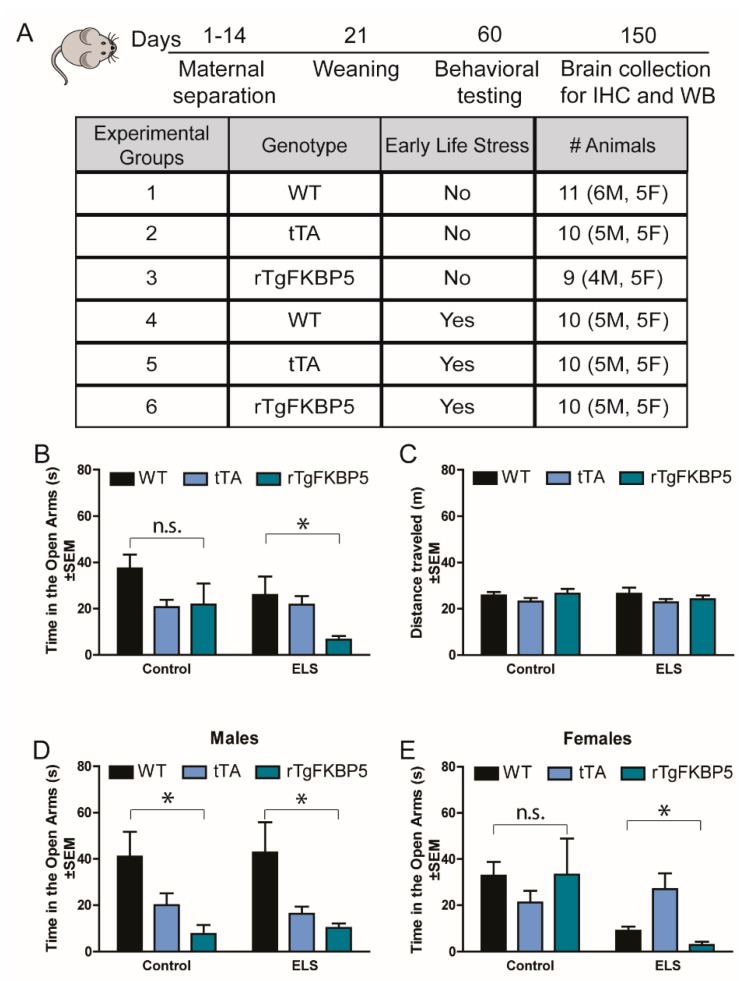
High FK506-binding protein 5 (FKBP5) and early life stress increases anxiety-like behavior. (**A**) Timeline of experimental protocol and total number of animals per group for each genotype and stress condition. For the maternal separation, mice were separated from their dams for 3 h daily for 14 consecutive days. (**B**) Total time in open arms and (**C**) distance traveled during the elevated plus maze (EPM) test. EPM total time spent in open arms by (**D**) males and (**E**) females. WT = wild-type, tTA = CamKIIα-tTA, rTgFKBP5 = FKBP5 overexpressing mice, M = males, F = Females, s = seconds, m =meters, ELS = early life stress. Data are represented as standard error of the mean (SEM) and analyzed by three- and two-way Analysis of variance (ANOVA)s (see Table 1 for analysis). Significant results were considered when *p* < 0.05 and were followed by Bonferroni post hoc tests. * = two-way ANOVA significant difference (*p* < 0.05) within non-stressed or ELS groups; n.s. = non-significant. MWM = Morris Water Maze.

**Figure 2 ijms-20-02738-f002:**
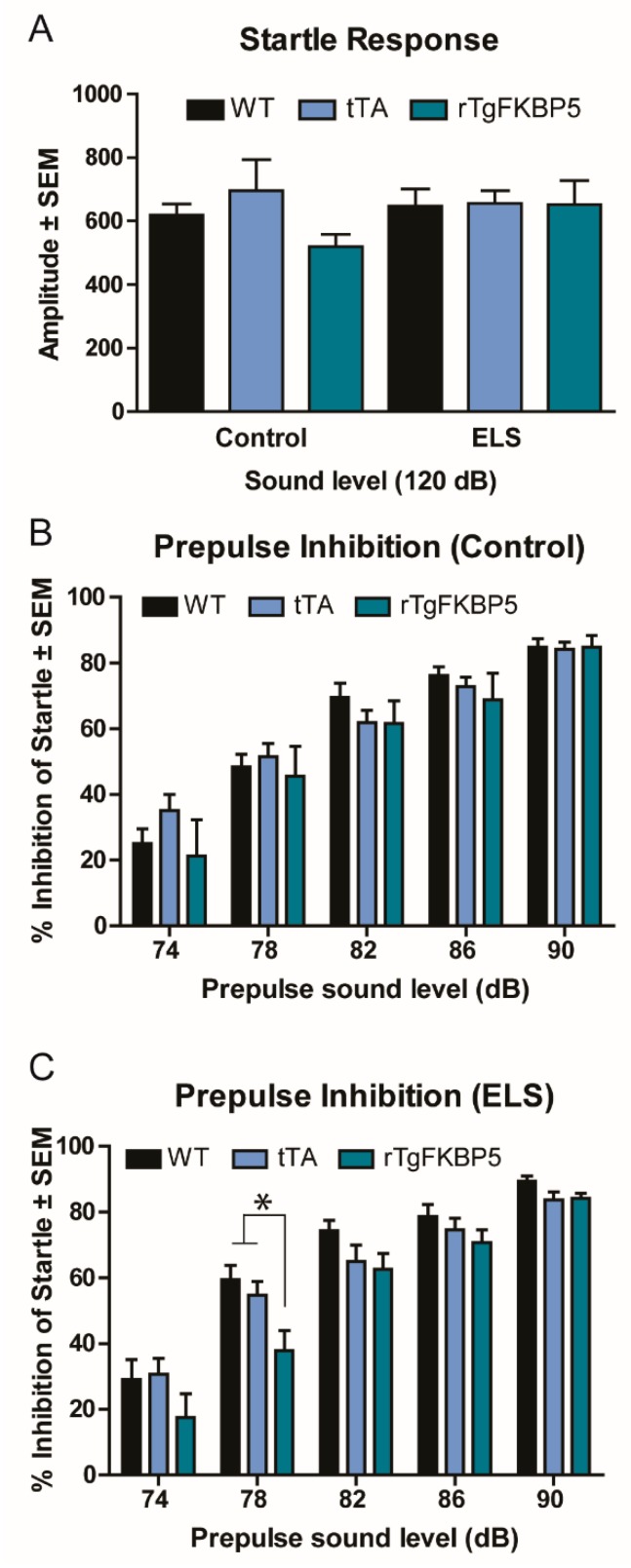
Interaction of FKBP5 and early life stress does not affect startle response. (**A**) Baseline startle response. Percentage of prepulse inhibition to 74, 78, 82, 86, and 90 dB acoustic stimulus compared to the baseline startle for the (**B**) control and (**C**) ELS groups. Data are represented as standard error of the mean (SEM) and analyzed by repeated measures two-way ANOVA (see Table 1 for analysis). * = *p* < 0.05 when comparing ELS-genotype differences. ELS = early life stress.

**Figure 3 ijms-20-02738-f003:**
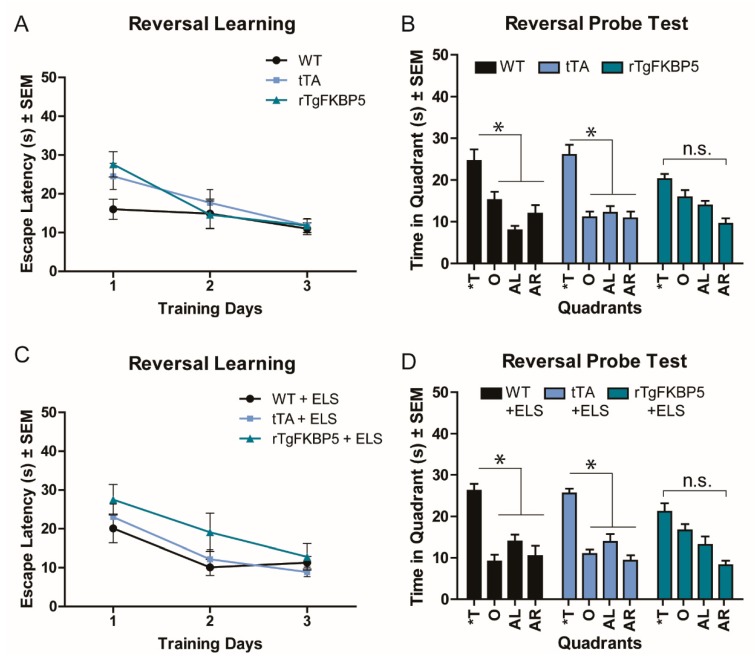
Deficit in spatial reversal learning is attributed to high levels of FKBP5 independently of stress. Spatial reversal memory analysis of non-stressed mice (WT= 11, tTA =10, TgFKBP5 = 9) during (**A**) training days and (**B**) probe test using the Morris Water Maze (MWM) paradigm. Data analysis for spatial reversal learning of early stressed mice (WT = 10, tTA = 10, rTgFKBP5 = 10) during (**C**) training days and (**D**) probe test. Each training day represents a block of 4–60 s trials. s = seconds, ELS = early life stress; Quadrants: T = target, O = Opposite, AR = adjacent right, AL = adjacent left. SEM = standard error of the mean, n.s. = non-significant. Data were analyzed by two-way ANOVA followed by Bonferroni post-hoc tests where significant results are represented as * = *p* < 0.05. Refer to Appendix A for MWM spatial learning and memory testing and Table 1 for three-way ANOVA analysis.

**Figure 4 ijms-20-02738-f004:**
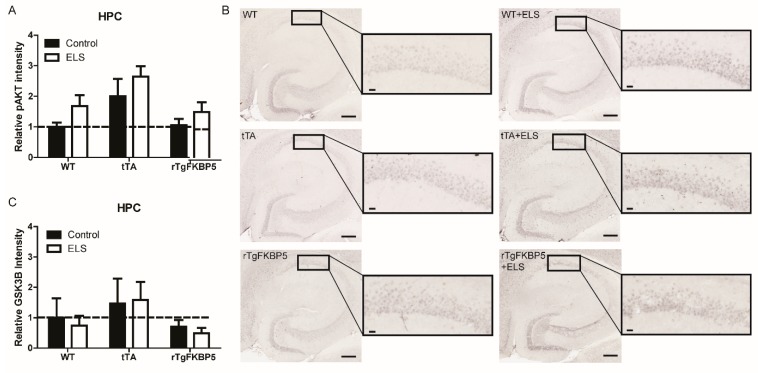
Early life stress increases phosphorylated AKT at Serine 473 (pAKT^Ser473^) in the hippocampus. (**A**) Relative intensity of pAKT^Ser473^ in the hippocampus (HPC) from WT = 6, tTA = 5, rTgFKBP5 = 5 mice and WT = 5, tTA = 5, rTgFKBP5 = 5 mice with ELS. (**B**) Representative images are shown. (**C**) GSK-3β levels in the HPC from the same mice. Interaction of stress and genotype was analyzed by two-way ANOVA followed by Bonferroni correction when significant (see Table 2). Values are represented as standard error of the mean (SEM) and relative protein expression is normalized to WT-control mice (represented by dotted lines). ELS = early life stress. Scale bar represents 200 µm; inset scale represents 20 µm. AKT = Protein kinase B.

**Figure 5 ijms-20-02738-f005:**
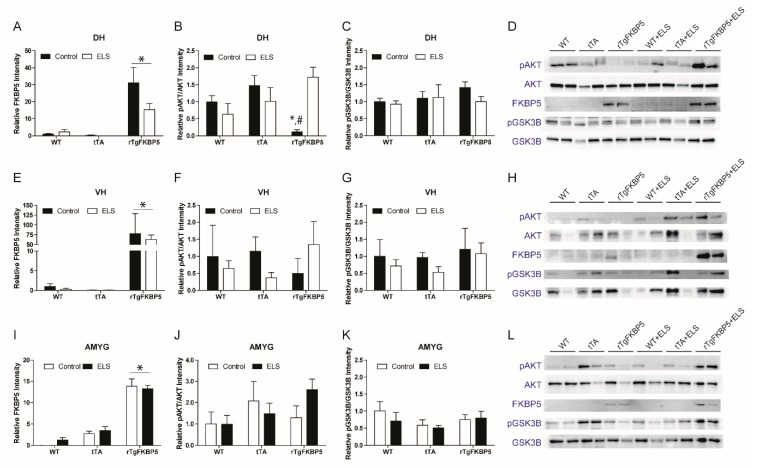
Overexpression of FKBP5 reduces pAKT/AKT ratio in the dorsal hippocampus. (**A**) FKBP5 protein expression from DH tissue punches normalized to total protein. Calculated ratio of (**B**) pAKT/AKT and (**C**) pGSK3β/GSK3β in DH. (**D**) Representative Western blots for DH. Protein expression calculated for (**E**) FKBP5, (**F**) pAKT/AKT and (**G**) pGSK3β/GSK3β ratios in the VH. (**H**) Representative Western blots for VH. Tissue punches were also extracted from the AMYG to assess protein expression of (**I**) FKBP5 followed by (**J**) pAKT/AKT and (**K**) pGSK3β/GSK3β in this structure. (**L**) Representative Western blots for AMYG. Interaction of stress and genotype was analyzed by two-way ANOVA followed by Bonferroni correction when significant * = *p* < 0.05.; # = significant difference between stressed and non-stressed rTgFKBP5 mice. Values are represented as standard error of the mean (SEM) and protein was normalized to total protein expression. DH = dorsal hippocampus, VH= ventral hippocampus, AMYG = amygdala, ELS = early life stress.

**Table 1 ijms-20-02738-t001:** Summary of statistical analyses of behavioral tasks.

Figure	Measured Condition	Groups Analyzed	Gen. x ELS	Factor	F Statistic and *p*-Value
1B	EPM: Anxiety levels (open arms) *See 1D-E for sex differences*	3-way ANOVA, All groups	F_(2, 48)_ = 1.078, *p* = 0.347	**Genotype**	F_(2, 48)_ = 4.610, *p* = 0.014
				**ELS**	F_(1, 48)_ = 3.28, *p* = 0.075
				**Sex**	F_(1, 48)_ = 0.308, *p* = 0.582
				**Sex/Gen/ELS**	F_(2, 48)_ = 1.968, *p* = 0.151
		2-way ANOVA, Non-stressed	F_(2, 48)_ = 1.548, *p* = 0.233	**Genotype**	F_(2, 48)_ = 1.442, *p* = 0.256
				**Sex**	F_(1, 48)_ = 0.617, *p* = 0.439
		2-way ANOVA, ELS	F_(2, 48)_ = 3.089, *p* = 0.06	**Genotype**	F_(2, 48)_ = 4.015, *p* = 0.031
				**Sex**	F_(1, 48)_ = 4.003, *p* = 0.05
	Number entries to open arms	3-way ANOVA, All groups	F_(2, 48)_ = 0.351, *p* = 0.705	**Genotype**	F_(2, 48)_ = 1.283, *p* = 0.285
				**ELS**	F_(1, 48)_ = 0.049, *p* = 0.824
				**Sex**	F_(1, 48)_ = 1.478, *p* = 0.230
				**Sex/Gen/ELS**	F_(1, 48)_ = 0.541, *p* = 0.585
	Anxiety levels (closed arms)	3-way ANOVA, All groups	F_(2, 48)_ = 1.181, *p* = 0.316	**Genotype**	F_(2, 48)_ = 0.010, *p* = 0.989
				**ELS**	F_(1, 48)_ = 0.850, *p* = 0.360
				**Sex**	F_(1, 48)_ = 0.638, *p* = 0.428
				**Sex/Gen/ELS**	F_(2, 48)_ = 1.096, *p* = 0.342
		2-way ANOVA, Males	F_(2,24)_ = 0.228, *p* = 0.797	**Genotype**	F_(2,24)_ = 3.509, *p* = 0.049
				**ELS**	F_(1,24)_ = 0.002, *p* = 0.958
		2-way ANOVA, Females	F_(2,24)_ = 1.865, *p* = 0.176	**Genotype**	F_(2,24)_ = 1.649, *p* = 0.213
				**ELS**	F_(1,24)_ = 0.354, *p* = 0.557
1C	EPM: Locomotion	3-way ANOVA, All groups	F_(2,48)_ = 0.353, *p* = 0.704	**Genotype**	F_(2,48)_ = 1.657, *p* = 0.200
				**ELS**	F_(1,48)_ = 0.205 *p* = 0.652
				**Sex**	F_(1,48)_ = 0.399, *p* = 0.673
				**Sex/Gen/ELS**	F_(2,24)_ = 1.649, *p* = 0.213
		2-way ANOVA, Males	F_(2,24)_ = 0.137, *p* = 0.879	**Genotype**	F_(2,24)_ = 0.714, *p* = 0.499
				**ELS**	F_(1,24)_ = 0.465 *p* = 0.501
		2-way ANOVA, Females	F_(2,24)_ = 0.336, *p* = 0.717	**Genotype**	F_(2,24)_ = 1.327, *p* = 0.284
				**ELS**	F_(1,24)_ = 0.0 *p* = 1
1D	Anxiety levels (open arms)	2-way ANOVA, Males	F_(2,24)_ = 0.089, *p* = 0.914	**Genotype**	F_(2,24)_ = 8.900, *p* = 0.001
				**ELS**	F_(1,24)_ = 0.001, *p* = 0.971
1E		2-way ANOVA, Females	F_(2,24)_ = 3.069, *p* = 0.065	**Genotype**	F_(2,24)_ = 0.747 *p* = 0.294
				**ELS**	F_(1,24)_ = 6.449, *p* = 0.018
2A	Startle response	3-way ANOVA, All groups	F_(2, 48)_ = 0.997, *p* = 0.377	**Genotype**	F_(2, 48)_ = 1.058 *p* = 0.355
				**ELS**	F_(1, 48)_ = 0.609, *p* = 0.439
				**Sex**	F_(1, 48)_ = 0.388, *p* = 0.536
				**Sex/Gen/ELS**	F_(1, 48)_ = 0.395, *p* = 0.676
		2-way ANOVA, Males	F_(2,24)_ = 0.691, *p* = 0.511	**Genotype**	F_(2,24)_ = 0.770, *p* = 0.475
				**ELS**	F_(1,24)_ = 0.199, *p* = 0.660
		2-way ANOVA, Females	F_(2,24)_ = 0.464, *p* = 0.634	**Genotype**	F_(2,24)_ = 0.791, *p* = 0.464
				**ELS**	F_(1,24)_ = 0.789, *p* = 0.383
2B,C	PPI	3-way RM-ANOVA, All groups	F_(2, 48)_ = 0.629, *p* = 0.537	**Genotype**	F_(2, 48)_ = 7.658, *p* = 0.001
				**ELS**	F_(1, 48)_ = 0.272, *p* = 0.605
				**Sex**	F_(1, 48)_ = 4.532, *p* = 0.038
				**Sex/Gen/ELS**	F_(1, 48)_ = 0.620, *p* = 0.542
3A,C	MWM, Reversal training	3-way RM-ANOVA, All groups	F_(2, 54)_ = 0.688, *p* = 0.507	**Genotype**	F_(2, 54)_ = 2.685, *p* = 0.078
				**ELS**	F_(1, 54)_ = 0.123, *p* = 0.727
				**Sex**	F_(1, 54)_ = 0.021, *p* = 0.886
				**Sex/Gen/ELS**	F_(2, 54)_ = 0.385, *p* = 0.683
		2-way RM-ANOVA, Non-Stressed		**Genotype**	F_(2, 24)_ = 1.456, *p* = 0.253
				**Sex**	F_(1, 24)_ = 0.101, *p* = 0.753
				**Training**	F_(2, 23)_ = 28.02, *p* < 0.001
		2-way RM-ANOVA, ELS		**Genotype**	F_(2, 24)_ = 1.913, *p* = 0.169
				**Sex**	F_(1, 24)_ = 0.006, *p* = 0.941
				**Training**	F_(2, 23)_ = 17.53, *p* < 0.001
		2-way ANOVA, Males	F_(2,24)_ = 0.812, *p* = 0.456	**Genotype**	F_(2,24)_ = 3.454, *p* = 0.048
				**ELS**	F_(1,24)_ = 0.003, *p* = 0.456
		2-way ANOVA, Females	F_(2,24)_ = 0.054, *p* = 0.947	**Genotype**	F_(2,24)_ = 0.091, *p* = 0.913
				**ELS**	F_(1,24)_ = 0.245, *p* = 0.625
3B,D	MWM, Reversal Probe	3-way RM-ANOVA, All groups	F_(2, 54)_ = 0.135, *p* = 0.873	**Genotype**	F_(2, 54)_ = 3.827, *p* = 0.027
				**ELS**	F_(1, 54)_ = 0.170, *p* = 0.681
				**Sex**	F_(1, 54)_ = 0.183, *p* = 0.670
				**Sex/Gen/ELS**	F_(2, 54)_ = 0.244, *p* = 0.784
		2-way ANOVA, Males	F_(2,24)_ = 0.076, *p* = 0.927	**Genotype**	F_(2,24)_ = 1.208, *p* = 0.316
				**ELS**	F_(1,24)_ = 0.712, *p* = 0.407
		2-way ANOVA, Females	F_(2,24)_ = 0.289, *p* = 0.752	**Genotype**	F_(2,24)_ = 0.012, *p* = 0.988
				**ELS**	F_(1,24)_ = 2.547, *p* = 0.124
S1A	Percent of open arm entries	3-way ANOVA, All groups	F_(2, 48)_ = 1.560, *p* = 0.219	**Genotype**	F_(2, 48)_ = 3.956, *p* = 0.026
				**ELS**	F_(1, 48)_ = 0.000, *p* = 0.998
				**Sex**	F_(1, 48)_ = 0.348, *p* = 0.558
				**Sex/Gen/ELS**	F_(1, 48)_ = 1.720, *p* = 0.097
S1B	Percent of open arm entries	2-way ANOVA, Males	F_(2, 48)_ = 2.753, *p* = 0.042	**Genotype**	F_(2, 48)_ = 5.423, *p* = 0.01
				**ELS**	F_(1, 48)_ = 1.493, *p* = 0.234
S1C	Percent of open arm entries	2-way ANOVA, Females	F_(2, 48)_ =0.854, *p* = 0.526	**Genotype**	F_(2, 48)_ = 0.199, *p* = 0.821
				**ELS**	F_(1, 48)_ = 1.627, *p* = 0.214
S2A		2-way RM-ANOVA, Males	F_(2,24)_ = 0.412, *p* = 0.668	**Genotype**	F_(2,24)_ = 0.614, *p* = 0.552
				**ELS**	F_(1,24)_ = 0.879, *p* = 0.360
S2B		2-way RM-ANOVA, Females	F_(2,24)_ = 7.092, *p* = 0.004	**Genotype**	F_(2,24)_ = 0.692, *p* = 0.511
				**ELS**	F_(1,24)_ = 3.903, *p* = 0.061
S3A	MWM, Visible	3-way ANOVA, All groups	F_(2, 54)_ = 0.107, *p* = 0.898	**Genotype**	F_(2, 54)_ = 2.062, *p*= 0.137
				**ELS**	F_(1, 54)_ = 0.110, *p*= 0.740
				**Sex**	F_(1, 54)_ = 1.289, *p* = 0.262
				**Sex/Gen/ELS**	F_(2, 54)_ = 0.099, *p* = 0.906
		2-way ANOVA, Males	F_(2,24)_ = 0.352, *p* = 0.707	**Genotype**	F_(2,24)_ = 5.372, *p* = 0.012
				**ELS**	F_(1,24)_ = 0.488, *p* = 0.492
		2-way ANOVA, Females	F_(2,24)_ = 0.008, *p* = 0.992	**Genotype**	F_(2,24)_ = 0.370, *p* = 0.694
				**ELS**	F_(1,24)_ = 0.002, *p* = 0.961
S3B,D	MWM, Training	3-way RM-ANOVA, All groups	F_(2, 54)_ = 0.407, *p* = 0.668	**Genotype**	F_(2, 54)_ = 0.434, *p* = 0.650
				**ELS**	F_(1, 54)_ = 1.231, *p* = 0.272
				**Sex**	F_(1, 54)_ = 0.003, *p* = 0.954
				**Sex/Gen/ELS**	F_(2, 54)_ = 0.195, *p* = 0.823
		2-way RM-ANOVA, Non-Stressed		**Genotype**	F_(2, 24)_ = 0.020, *p* = 0.981
				**Sex**	F_(1, 24)_ = 0.053, *p* = 0.821
				**Training**	F_(2, 23)_ = 14.27, *p* < 0.001
		2-way RM-ANOVA, ELS		**Genotype**	F_(2, 24)_ = 0.847, *p* = 0.441
				**Sex**	F_(1, 24)_ = 0.028, *p* = 0.868
				**Training**	F_(2, 23)_ = 26.49, *p* < 0.001
		2-way ANOVA, Males	F_(2,24)_ = 0.174, *p* = 0.841	**Genotype**	F_(2,24)_ = 0.092, *p* = 0.912
				**ELS**	F_(1,24)_ = 0.838, *p* = 0.369
		2-way ANOVA, Females	F_(2,24)_ = 0.402, *p* = 0.673	**Genotype**	F_(2,24)_ = 0.534, *p* = 0.593
				**ELS**	F_(1,24)_ = 0.313, *p* = 0.581
S3C,E	MWM, Probe	3-way ANOVA, All groups	F_(2, 54)_ = 7.047, *p* = 0.002	**Genotype**	F_(2, 54)_ = 7.018, *p* = 0.002
				**ELS**	F_(1, 54)_ = 9.118, *p* = 0.004
				**Sex**	F_(1, 54)_ = 0.057, *p* = 0.813
				**Sex/Gen/ELS**	F_(1, 54)_ = 0.362, *p* = 0.698

ELS: Early life stress, GEN: Genotype.

**Table 2 ijms-20-02738-t002:** Summary of statistical analyses by two-way ANOVA of staining and Western blots. DH = dorsal; VH = ventral; AMYG = amygdala.

Figure	Brain Area	Protein	Groups	Gen. x ELS	Gen.	ELS
Figure 4A	HPC	pAKT^Ser473^	All groups	F_(2, 24)_ = 0.077, *p* = 0.926	F_(2, 24)_ = 5.903, *p* = 0.007	F_(1, 24)_ = 4.373, *p* = 0.046
Figure 4C	HPC	GSK-3β	All groups	F_(2, 24)_ = 0.079, *p* = 0.923	F_(2, 24)_ = 1.639, *p* = 0.215	F_(1, 24)_ = 0.083, *p* = 0.774
Figure 5A	DH	FKBP5	All groups	F_(2, 24)_ = 3.563, *p* = 0.045	F_(2, 24)_ = 27.23, *p* < 0.001	F_(1, 24)_ = 3.068, *p* = 0.093
Figure 5B	DH	pAKT^Ser473^/AKT ratio	All groups	F_(2, 22)_ = 6.304, *p* = 0.006	F_(2, 22)_ = 1.110, *p* < 0.347	F_(1, 22)_ = 0.965, *p* = 0.336
Figure 5C	DH	pGSK-3 β^Ser9^/GSK-3β	All groups	F_(2, 22)_ = 0.575 *p* = 0.570	F_(2, 22)_ = 0.703, *p* < 0.505	F_(1, 22)_ = 0.812, *p* = 0.376
Figure 5E	VH	FKBP5	All groups	F_(2, 22)_ = 0.898, *p* = 0.421	F_(2, 22)_ = 15.51, *p* < 0.001	F_(1, 22)_ = 1.041, *p* = 0.318
Figure 5F	VH	pAKT^Ser473^/AKT ratio	All groups	F_(2, 22)_ = 0.746, *p* = 0.487	F_(2, 22)_ = 0.057, *p* < 0.944	F_(1, 22)_ = 0.159, *p* = 0.694)
Figure 5G	VH	pGSK-3β^Ser9^/GSK-3β	All groups	F_(2, 22)_ = 0.185, *p* = 0.832	F_(2, 22)_ = 1.175, *p* < 0.329	F_(1, 22)_ = 1.219, *p* = 0.286
Figure 5I	AMYG	FKBP5	All groups	F_(2,18)_ = 1.485, *p* = 0.253	F_(2, 18)_ = 118.6, *p* < 0.001	F_(1, 18)_ = 0.645, *p* = 0.432
Figure 5J	AMYG	pAKT^Ser473^/AKT ratio	All groups	F_(2, 18)_ = 1.446, *p* = 0.263	F_(2, 18)_ = 2.502, *p* < 0.115	F_(1, 18)_ = 0.774, *p* = 0.391
Figure 5K	AMYG	pGSK-3β^Ser9^/GSK-3β	All groups	F_(2, 18)_ = 0.374, *p* = 0.693	F_(2, 18)_ = 1.322, *p* < 0.291	F_(1, 18)_ = 0.462, *p* = 0.505

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
