# Peer review of "Early Life Stress and High FKBP5 Interact to Increase Anxiety-Like Symptoms through Altered AKT Signaling in the Dorsal Hippocampus"

_ijms, 2019, doi:10.3390/ijms20112738_

Round 1

Reviewer 1 Report

The manuscript describes the interaction between genes and environment, trhough a series of well designed behavioral tests following early life stress. It further shows further  the existence of a link between development of anxiety and AKT signaling in the dorsal hippocampus. 

I have a some concern regarding the PTSD and Hsp90 as keywords, since they are not representative of the research's contents.

In the Abstract authors hypothesized that interactions between early life stress and high FKBP5 expression induce brain changes affecting cognitive and emotional processes. However, the results seems to argue against a disorder of the emotional reactivity and to indicate that, with involvement f the dorsal hippocampus, the cognitive aspect appears to be linked to the observed gene expression. Taking into account the functional difference between dorsal and ventral hippocampus, the discussion should point out how the study could be continued to understand how cognitive and emotional components may influence the maladaptive plasticity due to ELS. 

Discussion is long and complex and should benefit from division into subheadings.  

Author Response

Reviewer 1

The manuscript describes the interaction between genes and environment, through a series of well-designed behavioral tests following early life stress. It further shows further the existence of a link between the development of anxiety and AKT signaling in the dorsal hippocampus.

I have some concern regarding the PTSD and Hsp90 as keywords, since they are not representative of the research's contents.

1.      We agree and thank the reviewer for this observation. We have updated our keywords to remove PTSD and Hsp90 and include the following: “FKBP5, early life stress, anxiety, hippocampus, AKT” which now better represent the study’s content.

In the Abstract authors hypothesized that interactions between early life stress and high FKBP5 expression induce brain changes affecting cognitive and emotional processes. However, the results seem to argue against a disorder of the emotional reactivity and to indicate that, with involvement of the dorsal hippocampus, the cognitive aspect appears to be linked to the observed gene expression. Taking into account the functional difference between dorsal and ventral hippocampus, the discussion should point out how the study could be continued to understand how cognitive and emotional components may influence the maladaptive plasticity due to ELS.

2.      We thank the reviewer for providing this comment to strengthen our paper. We have updated the abstract and discussion. Specifically, we have now included our hypothesis in broader terms in the abstract, “We hypothesized that interactions between ELS and high FKBP5 induce phenotypic changes that correspond to underlying molecular changes in the brain.” In the discussion, we added the following information:  

“Besides sex differences, it will be interesting to further study how ELS differentially affect the hippocampal axes’ synaptic plasticity in rTgFKBP5 mice. This information could contribute to our knowledge on how cognitive and emotional components may influence hippocampal plasticity and activity after stressful events. These changes may also inform us about long-lasting neuronal changes affecting our susceptibility to stress at later ages.”

In the Concluding Remarks, we have now included mention of follow-up studies are important to better understand these aspects.

Discussion is long and complex and should benefit from division into subheadings. 

3.      Thank you for your constructive comment. We have revised the discussion, including cutting text that was not as directly linked to our results and added in subheadings, as you have suggested.

Reviewer 2 Report

In this manuscript Criado-Marrero and colleagues examined the interaction between FKPB5 over-expression and early life adversity on memory and anxiety-like behavioral endpoints. The authors also explored the consequences of FKPB5 over-expression on signaling pathways that are related to the stress response, FKBP5 function and memory or anxiety-like behavior. The results here add to an existing body of literature on the impact of FKPB5 and early life adversity on behavior and biology. The studies are well designed and the results are interesting. 

The authors should be lauded for two important components of their experimental design: 1) they included WT and tTA only animals as controls. The tTA only mice represent a very important control for this study.

2) The authors studied the effects of FKPB5 over-expression and ELS in both males and females. 

However, there are major concerns regarding some of the methodology for statistical analyses. Furthermore, the presentation of some of the results is confusing and there are concerns with the some of the interpretation of the data. Details are below.

Tables 1 and 2 show that the authors used ANOVAs for the majority of the analyses, which is appropriate. However, some of the specific results are missing. Some of the ANOVAs were not appropriate for the specific question at hand and some of the post-hoc comparisons are either unclear or incorrect in more than one of the experiments. 

Figure 1B. The authors do not report the sex by genotype by ELS interaction. It also seems that the authors chose to use unpaired t-tests rather than post hoc tests such as Dunnett or Bonferroni to campare across conditions and sex. Fig. 1B "control": the WT seem higher than both TtA AND rTgFKBP5 and I could not find that comparison reported anywhere in the manuscript.

For the Non-Stressed and ELS portions of the analyses, the authors took out the "sex" factor and it is not clear why that is (still in the upper part of Table 1). It would be helpful if all of the post-hoc tests performed were clearly labeled and reported, either in the results and/or as part of Table 1. For each condition, the authors should have performed a two way ANOVA with sex and genotype as between subject factors- rather than one-way ANOVAs reported here. 

Generally in Table 1, many of the interactions that should result from the analyses are not reported. The authors should rely on post-hoc tests rather than unpaired t-tests to make comparisons between groups. 

Figure 4.  The authors do not address the observation that genotype has an overall effect on pAKT phosphorylation. There are no symbols to depict the statistically significant changes reported by the ANOVAs. It appears that tTA mice have higher AKT phosphorylation regardless of condition, which is problematic. As described, it is very difficult to interpret the results of this figure. 

Overall, without the proper analysis and representation of these data, it is difficult to further evaluate the validity of the findings reported here. Some key points are also missing from the discussion, including addressing some of the discrepancies between behavioral endpoints and biochemical correlates in the brain shown in this manuscript. 

minor points:

In the introduction, more rationale could be provided for studying AKT signaling. 

The elevated plus maze has been shown to be problematic in females (see Fernandes et al. Pharmacol Biochem Behav, 1999). Generally speaking, anxiety-like behavior is difficult to assess in rodents and the authors only used one behavioral test in this study. This should be acknowledged in the discussion and the authors should mention that further tests would be valuable to confirm the findings reported here. 

Author Response

Reviewer 2

In this manuscript Criado-Marrero and colleagues examined the interaction between FKPB5 over-expression and early life adversity on memory and anxiety-like behavioral endpoints. The authors also explored the consequences of FKPB5 over-expression on signaling pathways that are related to the stress response, FKBP5 function and memory or anxiety-like behavior. The results here add to an existing body of literature on the impact of FKPB5 and early life adversity on behavior and biology. The studies are well designed and the results are interesting.

The authors should be lauded for two important components of their experimental design: 1) they included WT and tTA only animals as controls. The tTA only mice represent a very important control for this study.

2) The authors studied the effects of FKPB5 over-expression and ELS in both males and females.

However, there are major concerns regarding some of the methodology for statistical analyses. Furthermore, the presentation of some of the results is confusing and there are concerns with the some of the interpretation of the data. Details are below.

Tables 1 and 2 show that the authors used ANOVAs for the majority of the analyses, which is appropriate. However, some of the specific results are missing. Some of the ANOVAs were not appropriate for the specific question at hand and some of the post-hoc comparisons are either unclear or incorrect in more than one of the experiments.

We apologize for any confusion in the presentation and methodology of statistical analyses.  We have added additional analyses into the “Results” section and Table 1 and 2, including posthoc analysis, where appropriate. We hope you will find that this has clarified and corrected any previous issues with our analysis.

Figure 1B. The authors do not report the sex by genotype by ELS interaction. It also seems that the authors chose to use unpaired t-tests rather than post hoc tests such as Dunnett or Bonferroni to compare across conditions and sex. Fig. 1B "control": the WT seem higher than both tTA AND rTgFKBP5 and I could not find that comparison reported anywhere in the manuscript.

In Table 1: We report the influence of “Sex” as a factor and the “Sex/Gen/ELS interaction” [F(2, 48) = 1.968, p = 0.151] obtained from the Three-way ANOVA. We also confirm that despite that WT seem higher than both tTA AND rTgFKBP5 in the control group, they did not show significance in the Bonferroni correction test.

In the results, we also include the results from the Bonferroni post-hoc test and the explanation to follow this with an independent sample t-test.

Although the genotype presented a significant effect on the ELS group (Table 1), the number of factors being studied reduced the significant power in the Bonferroni post-hoc test (p > 0.05). Additionally, sex significantly contributed to the variability (p = 0.05) in the ELS group. However, due the small representation of animals per sex on each group, we ad-hoc an independent sample t-test comparing the groups. In this analysis, the rTgFKBP5-ELS mice displayed increased anxiety-like behavior as measured by a significant decrease in time in the open arms compared to tTA-ELS (t(18) = 3.489, p = 0.002) and WT-ELS (t(18) = 2.261, p = 0.036) mice (Figure 1B), which was not affected by the number of entries to open arms (Table 1) or locomotor differences among the groups (Figure 1C).”

For the Non-Stressed and ELS portions of the analyses, the authors took out the "sex" factor and it is not clear why that is (still in the upper part of Table 1). It would be helpful if all of the post-hoc tests performed were clearly labeled and reported, either in the results and/or as part of Table 1. For each condition, the authors should have performed a two way ANOVA with sex and genotype as between subject factors- rather than one-way ANOVAs reported here.

As indicated above, we have included “sex” factor in Table 1. Bonferroni post-hoc tests were performed after any   significant p-value was obtained from three- or two-way ANOVAs. For better clarity, these values were included in the text when discussing their corresponding findings. As suggested, we performed two-way ANOVAs for each condition using both sex and genotype as factors. 

Generally in Table 1, many of the interactions that should result from the analyses are not reported. The authors should rely on post-hoc tests rather than unpaired t-tests to make comparisons between groups.

Thank you for pointing this out, we have reformatted Table 1 including all the interactions and information for better understanding of our results. We’ve added the post-hoc analysis, where appropriate, into the results section.

Figure 4.  The authors do not address the observation that genotype has an overall effect on pAKT phosphorylation. There are no symbols to depict the statistically significant changes reported by the ANOVAs. It appears that tTA mice have higher AKT phosphorylation regardless of condition, which is problematic. As described, it is very difficult to interpret the results of this figure.

We agree that the data does appear to show an increase in signal for the tTA group, however, this is not statistically significant from the other groups. We have now acknowledged this non-significant increase in the discussion and added one possible rationale, that the levels of CamKIIα in the tTA mice may be affected by the transgene, and this may cause downstream processes to be affected including AKT, which has been previously linked (PMID: 19545622).

Overall, without the proper analysis and representation of these data, it is difficult to further evaluate the validity of the findings reported here. Some key points are also missing from the discussion, including addressing some of the discrepancies between behavioral endpoints and biochemical correlates in the brain shown in this manuscript.

We apologize for the confusion on the presentation of the statistical analyses and the missing information in the discussion. In line with your recommendation, we added more explanation about the discrepancies between behavior and biochemical findings in our study. Since we found that sex greatly influence the behavioral responsiveness, we discuss this as a limitation to make specific conclusions about cognitive and behavioral correlations from the FKBP5 and ELS interaction in our study. We added the following information in our discussion:

“One limitation of our study is lacking an appropriate number of animals representing each sex per each group which prevent us from making conclusions about possible sex influence on molecular findings. We will also need this information to establish any possible correlation between behavior and molecular outcomes.”

minor points:

In the introduction, more rationale could be provided for studying AKT signaling.

Thank you for your suggestion. We have added a short rationale of studying AKT signaling in the introduction.

“We also examined AKT phosphorylation, since it has been previously demonstrated to be inhibited by FKBP5 [18] and is known to regulate autophagy, cell survival, and hippocampal synaptic plasticity affecting learning and memory processes [19,20].”

 The elevated plus maze has been shown to be problematic infemales (see Fernandes et al. Pharmacol Biochem Behav, 1999). Generally speaking, anxiety-like behavior is difficult to assess in rodents and the authors only used one behavioral test in this study. This should be acknowledged in the discussion and the authors should mention that further tests would be valuable to confirm the findings reported here.

Thank you for pointing this out and we agree that additional testing will be important in follow-up studies to confirm our results here. We have added this observation in our discussion:

“It is important to note that previous studies have demonstrated significant differences in female rats using EPM (PMID: 10593196), which may be applicable to other rodents. It will be valuable to confirm these using other anxiety-like behavioral tests in future studies. Anxiety-like behaviors, as many other behaviors, are difficult to assess in rodents. This difficulty arises from the number of mice needed to observe significant differences, the inclusion of sex as an independent variable in previous studies to make valuable comparisons, and the limited information about the biochemical and physiological differences between sexes in rodents’ brain.”

Reviewer 3 Report

Having read the manuscript entitled “Early life stress and high FKBP5 interact to increase anxiety-like symptoms through altered AKT signaling in the dorsal hippocampus” I have to admit that the topic is quite interesting. However, I have a few comments:

1. The Introduction section should not contain information what was observed during the presented experiments. Such information should be given in the Discussion/Conclusion section.

2. The Results section is very long and confusing, since it contains a lot of information that should be given in the Methods and materials or Discussion sections. The Results section should refer only to the obtained outcomes, without discussing them or explaining the used methods. Therefore, the Result section should be re-written.

3. In the Result section the Authors should also present the percentage of open arm entries.

4. Line 120 – results related only to open arms (not closed arms) were presented.

5. Line 125 – Fig 1B presents a significant decrease in time spent in the open arms only vs. WT-ELS group.

6. Line 209 – Differences between groups are not indicated in Fig. 4A.

7. In the Introduction or Discussion section the Authors should explain why the tTA control group was used.

Author Response

Reviewer 3

Having read the manuscript entitled “Early life stress and high FKBP5 interact to increase anxiety-like symptoms through altered AKT signaling in the dorsal hippocampus” I have to admit that the topic is quite interesting. However, I have a few comments:

1. The Introduction section should not contain information what was observed during the presented experiments. Such information should be given in the Discussion/Conclusion section.

We have removed extraneous information describing the current study from the introduction. However, we have maintained the final paragraph of the introduction as a summary of the current study, as is typical in scientific publications.

2. The Results section is very long and confusing, since it contains a lot of information that should be given in the Methods and materials or Discussion sections. The Results section should refer only to the obtained outcomes, without discussing them or explaining the used methods. Therefore, the Result section should be re-written.

Thank you for your comment. We have revised the results, methods, and discussion sections to improve clarity and readability by removing and reordering content. 

3. In the Result section the Authors should also present the percentage of open arm entries.

Thank you for pointing this out. We have added the percentage of open arm entries into the Supplementary File, which is now also described in the results.

4. Line 120 – results related only to open arms (not closed arms) were presented.

We have added the data for closed arm time into Table 1.

5. Line 125 – Fig 1B presents a significant decrease in time spent in the open arms only vs. WT-ELS group.

We thank the reviewer for this comment. We have included the statistical analysis in Table 1 and addressed this observation in the figure by including the non-significant labeling and explaining it in the “Results”

6. Line 209 – Differences between groups are not indicated in Fig. 4A.

Thank you for your comment. The significance was by ELS overall by two-way ANOVA. We think adding this into the graph may increase confusion. Please advise if there is a specific way you recommend denoting this, that we would be happy to include.

7. In the Introduction or Discussion section the Authors should explain why the tTA control group was used.

Thank you for letting us know that this would be helpful. The tTA group is an essential control because it is a part of the background of the rTgFKBP5 mice, which is made from crossing an FVB-FKBP5 mouse with a CamKIIα-tTA mouse. We have now included additional information in the Results “We used both the WT and tTA controls to ensure any effect of high FKBP51 was independent of either background” and Discussion “We used a unique mouse model overexpressing FKBP5 (rTgFKBP5) and two control groups (WT and tTA) to address our hypothesis. The tTA control group express the tetracycline-off transactivator driven by the CAMKII promoter which helped us distinguish rTgFKBP5 mice from any phenotypical and biochemical difference caused by this construct.”

Round 2

Reviewer 1 Report

In the revised manuscript authors have accomplished all the reviewer's concerns. Still present some English minor spelling mistakes. Overall, the manuscript has noticeably improved and I recommend now publication. 

Reviewer 2 Report

The authors have addressed my comments

Reviewer 3 Report

All my comments have been addressed.